# Use of AbobotulinumtoxinA in Adults with Cervical Dystonia: A Systematic Literature Review

**DOI:** 10.3390/toxins12080470

**Published:** 2020-07-24

**Authors:** Alfonso Fasano, Vijayashankar Paramanandam, Mandar Jog

**Affiliations:** 1Edmond J. Safra Program in Parkinson’s Disease, Morton and Gloria Shulman Movement Disorders Clinic, Toronto Western Hospital, UHN, Toronto, ON M5T2S8, Canada; drvijayashankar@gmail.com; 2Division of Neurology, University of Toronto, Toronto, ON M5S 3H2, Canada; 3Krembil Brain Institute, Toronto, ON M5T 1M8, Canada; 4Lawson Health Research Institute, London, ON N6A 4V2, Canada; Mandar.Jog@lhsc.on.ca

**Keywords:** abobotulinumtoxinA, cervical dystonia, Dysport, systematic literature review, treatment

## Abstract

Cervical dystonia (CD) is a neurological movement disorder characterized by sustained involuntary muscle contractions. First-line therapy for CD is intramuscular injections of botulinum neurotoxin (e.g., abobotulinumtoxinA) into the affected muscles. The objective of this systematic literature review is to assess the clinical evidence regarding the effects of abobotulinumtoxinA for treatment of CD in studies of safety, efficacy, patient-reported outcomes, and economic outcomes. Using comprehensive electronic medical literature databases, a search strategy was developed using a combination of Medical Subject Heading terms and keywords. Results were reviewed by two independent reviewers who rated the level of evidence. The search yielded 263 publications, of which 232 were excluded for being duplicate publications, not meeting the selection criteria, or failing to meet predefined eligibility criteria, leaving a total of 31 articles. Clinical efficacy, patient-reported outcomes, and safety data were in 6 placebo-controlled trials (8 articles), 6 active-controlled trials, and 16 observational studies (17 articles). Data on health economic outcomes were provided in one of the clinical trials, in two of the observational studies, and in one specific cost-analysis publication. This review demonstrated that the routine use of abobotulinumtoxinA in CD is well-established, effective, and generally well-tolerated, with a relatively low cost of treatment.

## 1. Introduction

Cervical dystonia (CD) is a chronic neurological movement disorder characterized by sustained involuntary muscle contractions, which frequently leads to abnormal head movements and disabling, sometimes painful, postures of the head and cervical spine [1]. The pathophysiology of CD is not well understood. CD primarily occurs in individuals aged 40‒50 years and the majority of cases are idiopathic, although about 10% of patients have a positive family history. Acquired CD is rare and can arise after exposure to anti-dopaminergic drugs, brain injury, and other neurological disorders, including neurodegenerative diseases (e.g., Parkinson’s disease). Worldwide, the estimated prevalence of CD varies anywhere from 20 to 4100 cases per million [2]. Estimates may underrepresent the true prevalence because of underdiagnosis or misdiagnosis.

The first-line symptomatic treatment of CD is intramuscular injections of botulinum neurotoxin into the affected muscles, which inhibits the release of acetylcholine at the neuromuscular junction [3,4]. In Western countries, there are three commercially available formulations of botulinum neurotoxin type A—onabotulinumtoxinA (Botox, Allergan), abobotulinumtoxinA (Dysport, Ipsen), and incobotulinumtoxinA (Xeomin, Merz Pharmaceuticals)—and one commercially available formulation of botulinum neurotoxin type B—rimabotulinumtoxinB (MyoBloc, Solstice Neurosciences/NeuroBloc, Eisai Co., Ltd.)—that are indicated for CD. The US Food and Drug Administration approved Dysport for the treatment of adults with CD in 2009 and Health Canada approved Dysport Therapeutic (abobotulinumtoxinA) for the treatment of adults with CD in 2017.

The aim of the current systematic literature review is to assess the depth of clinical evidence regarding the effects of abobotulinumtoxinA for the treatment of adults with CD in studies of safety, efficacy, patient-reported outcomes, and economic outcomes. Results from this analysis may inform the design of new clinical trial programs and provide an evidence-based resource for clinical practice.

## 2. Methods

Studies reporting on the effects of abobotulinumtoxinA for the treatment of adults with CD in terms of safety, efficacy, patient-reported outcomes, and economic outcomes were identified by way of a comprehensive systematic literature review performed in accordance with PRISMA (Prevention and Recovery Information System for Monitoring and Analysis) guidelines [5]. No language, publication date, or publication status restrictions were imposed. Three comprehensive electronic medical literature databases were searched (PubMed, Cochrane Library, and Embase) through 9 April 2018. The literature search strategy was developed using a combination of Medical Subject Heading (MeSH) terms and keywords. Relevant keywords and search strings are presented in Appendix A.

Search results were screened using the following inclusion criteria: (i) the study was interventional or observational in design; (ii) study patients were adults (aged ≥ 18 years) with CD; (iii) the therapeutic intervention included treatment with abobotulinumtoxinA; and (iv) eligible clinical trials had a control intervention such as placebo, onabotulinumtoxinA, incobotulinumtoxinA, rimabotulinumtoxinB, or Lanzhou BTX-A.

Two reviewers independently extracted information from the articles, first by reviewing titles and abstracts and then by reviewing full texts. Relevant information regarding (i) study type, (ii) number of patients and type of interventions used in the study, and (iii) outcomes and parameters utilized or outcome assessment was recorded. The reviewers rated the level of evidence and assessed bias using the Grading of Recommendations, Assessment, Development and Evaluations (GRADE) approach (Appendix A) [6].

## 3. Results

### 3.1. Identified Studies and Quality Assessment

A flowchart of the systematic literature search is shown in Figure 1. The search yielded a total of 263 potentially relevant publications across the three databases. On initial review, 77 articles were excluded for being duplicate publications. On title and abstract screening, an additional 116 publications were excluded either for not meeting the selection criteria or for being duplicates (e.g., a conference abstract being published as a full text). Lastly, an additional 39 articles were excluded on full text review for failing to meet the predefined eligibility criteria. Thus, a total of 31 articles were included in the analysis. Clinical efficacy, patient-reported outcomes, and safety data were from 6 placebo-controlled trials (covered by 8 publications; Table 1) [7,8,9,10,11,12,13,14], 6 active-controlled trials (Table 2) [15,16,17,18,19,20], and 16 observational studies (covered by 17 publications; Table 3) [21,22,23,24,25,26,27,28,29,30,31,32,33,34,35,36,37]. Data on health economic outcomes was provided for one of the clinical trials (but in a separate publication), in two of the observational studies, and in one specific cost-analysis publication.

### 3.2. Efficacy

Of 31 identified articles, a total of 26 publications published in the 1995–2018 period described clinical efficacy. Six studies were placebo-controlled, six active-controlled, two open-label, two prospective observational, one retrospective observational, two retrospective cohort, one longitudinal cohort, one database study, four miscellaneous prospective, and one retrospective chart review. Objective efficacy measures included the Toronto Western Spasmodic Torticollis Rating Scale (TWSTRS) [38,39] and Tsui scores [38]. Data from placebo-controlled studies are summarized in Table 1, active-controlled studies in Table 2, and observational studies in Table 3.

#### 3.2.1. Placebo-Controlled Studies

Dosing and injection pattern varied across the six placebo-controlled studies. Most studies used an average abobotulinumtoxinA dose of 500 U; a few studies used lower or higher doses (range, 125 to 1000 U; Table 1).

Poewe et al. (1998) demonstrated significantly greater reductions in Tsui score from baseline to week 4 with abobotulinumtoxinA 500 U and 1000 U relative to placebo (*p* < 0.05) in a placebo-controlled study of three doses of abobotulinumtoxinA (250 U, 500 U, and 1000 U) conducted in 73 adults with CD [7]. Wissel et al. (2001) described significant reductions in TWSTRS score from baseline to weeks 4 and 8 (*p* = 0.001 and *p* = 0.002, respectively) with abobotulinumtoxinA 500 U versus placebo in another small study (*N* = 68) [8]. A similar result to Wissel et al. (2001) was observed by Truong et al. (2005; Table 1) [9]. Lew et al. (2018) assessed abobotulinumtoxinA at a mean dose of 452 U (median dose of 500 U) in 134 patients, which showed change from baseline in mean TWSTRS total score at week 4 compared with placebo [10]. A statistically significant difference from placebo was observed at week 2. Again, abobotulinumtoxinA was shown to be effective both in botulinum toxin-naïve and in previously botulinum toxin-treated patients. The efficacy was maintained during the open-label extension phase, in which patients were treated with abobotulinumtoxinA for up to four treatment cycles (mean dose, cycle 1, 502 U; cycle 2, 643 U; cycle 3, 716 U; and cycle 4, 776 U). The relative dose adjustment from onabotulinumtoxinA to abobotulinumtoxinA used in the study was 1:2.5.

In the largest randomized, placebo-controlled trial to date, 648 patients were assigned to abobotulinumtoxinA solution (500 U), abobotulinumtoxinA dry formulation (500 U), or placebo for the double-blind phase [11]. Both formulations of abobotulinumtoxinA demonstrated a statistically significant difference versus placebo in change from baseline to week 4 and observed up to week 12 in TWSTRS total score and subscale scores. A further post hoc analysis indicated that abobotulinumtoxinA solution was non-inferior to dry formulation [11].

#### 3.2.2. Active-Controlled Studies

Our search identified six active-controlled studies. Odergren et al. (1998) enrolled 73 patients and used a 3:1 bioequivalence ratio to evaluate the dose equivalence and efficacy of abobotulinumtoxinA (mean dose of 477 U) and onabotulinumtoxinA (mean dose of 152 U) [15]. Both groups showed improvements in Tsui score up to week 12. The difference between treatment groups in mean post-treatment Tsui scores was not statistically significant. Laubis-Herrmann et al. (2002) compared the efficacy of abobotulinumtoxinA at the recommended dose (500 U) and at a low dose (125 U) in 31 patients; both dose groups experienced statistically significant improvements in TWSTRS total score and two subscale scores from baseline to week 4 [16]. The difference between groups was not statistically significant for the TWSTRS total or subscales scores. In a crossover study by Ranoux et al. (2002), 54 enrolled patients were randomly assigned to receive onabotulinumtoxinA at the usually effective dose (defined as the dose at which a satisfactory response was achieved in the previous two treatments), abobotulinumtoxinA at a dose ratios of 1:3 and 1:4 (onabotulinumtoxinA:abobotulinumtoxinA) [17]. At week 4, statistically significant improvements from baseline in Tsui score and TWSTRS pain score were observed with the abobotulinumtoxinA treatment compared to the onabotulinumtoxinA treatment. The mean duration of action was 7 days longer with abobotulinumtoxinA 1:3 and 25 days longer with abobotulinumtoxinA 1:4 than with onabotulinumtoxinA (*p* = 0.58 and *p* = 0.02, respectively). Yun et al. (2015) compared onabotulinumtoxinA (mean dose, 144 U) and abobotulinumtoxinA (mean dose, 361 U) at a 1:2.5 treatment ratio in 94 patients, with a 4-week washout period between 16-week treatment cycles [18]. There were no statistically significant differences between groups in Tsui score or TWSTRS total and subscale scores from baseline to week 4. The results demonstrated that onabotulinumtoxinA was non-inferior to abobotulinumtoxinA at the 1:2.5 treatment ratio. Barbosa et al. (2015) compared Lanzhou BTX-A with abobotulinumtoxinA at an equivalency ratio of 1:3 U (actual doses are not reported) [19]. Both treatments also demonstrated statistically significant decreases from baseline in all TWSTRS subscale scores at week 4 after the initial injection, again with no statistically significant difference between the two treatments.

An open-label study of 28 patients by Van den Bergh et al. (1995) demonstrated improvement in the mean composite score (comprising subjective rating Tsui score and video score) in patients treated with abobotulinumtoxinA 384 U (mean composite score was 18.9 at baseline and improved to 5.2 at peak improvement) [21]. Mean composite scores dropped significantly from first treatment to last (mean: five cycles). In a study by Kessler et al. (1999) involving 303 patients who received at least six treatments with abobotulinumtoxinA, statistically significant reductions in modified Tsui score over the first six injections were observed and remained generally consistent up to the 15th treatment [22]. The greatest reduction in modified Tsui score was seen after the first treatment. Similar improvement in Tsui scores was noted with abobotulinumtoxinA in three other studies (Table 3) [23,24,26].

An expert group of neurologists observed 404 patients for treatment response following a single injection of botulinum toxin A: approximately 33% of patients treated with abobotulinumtoxinA (median dose, 500 U) and 23% of patients treated with onabotulinumtoxinA (median dose, 160 U) were considered treatment responders [27]. The ANCHOR-CD prospective registry study analyzed 304 patients and showed the TWSTRS total score decreased by 27.4% from baseline to week 4 after treatment with abobotulinumtoxinA (mean dose, 504 U), with a 31.7%, 18.5%, and 25.3% decrease in the TWSTRS severity, disability, and pain subscale scores, respectively [28].

Bentivoglio et al. (2017) assessed long-term efficacy and safety of abobotulinumtoxinA (mean dose, 702 U) in 39 patients who cumulatively received more than 750 total treatments [29]. Mean Tsui score was 5.7 before treatment and 3.5 at maximum efficacy. A ≥2-point reduction (improvement) in Tsui score was observed in 70.9% of the treatments. The mean overall duration of clinical improvement was 93 days and the median inter-treatment interval was 131 days. Hefter et al. (2014) analyzed 568 patients and reported that 5.8% of patients had developed partial secondary treatment failure (PSTF; determined using ≥4 Tsui scores collected during treatment with ≥3 consecutive abobotulinumtoxinA injections) [30]. Incidence of PSTF was estimated to be 1.6% per year. The time of onset of PSTF varied, with one patient experiencing PSTF after 4 injections and another after 38 injections.

#### 3.2.3. Patient-Reported Outcomes

Eighteen abobotulinumtoxinA studies conducted from 1995 to 2018 described patient-reported outcomes. Seven of these were placebo-controlled studies (covered by eight publications), two were active-controlled studies, two were pharmacoeconomic studies, two were prospective open-label studies, and two were longitudinal studies; one long-term and one real-world registry described patient-related outcomes. In total, five different tools were used to assess quality of life (QoL) in CD patients: the 36-Item Short Form Health Survey (SF-36) [12], the Patient Global Impression of Change (PGIC) [10,23], the Cervical Dystonia Impact Profile-58 (CDIP-58) [10,11], the Craniocervical Dystonia Questionnaire-24 (CDQ-24) [24,25], and the Visual Analog Scale (VAS) [9,11,12,24,25,29], as well as study-specific subjective measures of change and duration of effect.

Poewe et al. (1998) demonstrated a subjective global improvement in modified Tsui scale rating of >50% for the 1000 U-treated group at weeks 4 and 8 and for the 500 U-treated group at week 8 [11]. Wissel et al. (2001) observed that patients treated with abobotulinumtoxinA were 3.3 times more likely than those randomly assigned to placebo to experience a reduction in CD-associated pain at week 4, and were 8.5 times more likely than those receiving placebo to indicate a global improvement of disease at week 4 and 6.8 times more likely to do so at week 8 [8]. Truong et al. (2005) demonstrated a statistically significant improvement in patient-assessed change in CD signs and symptoms with abobotulinumtoxinA versus placebo at weeks 4 and 8 [9].

Laubis-Herrmann et al. (2002) observed 77% of patients who had received the recommended dose of abobotulinumtoxinA indicated “marked improvement” and 23% indicated “mild improvement,” compared with 50% and 29%, respectively, in the low-dose group [16]. In another study, 14% of patients with abobotulinumtoxinA and 25% of patients with Lanzhou BTX-A reported the improvement persisted for more than three months after the first injection. After the fifth treatment, 55% with abobotulinumtoxinA and 50% with Lanzhou BTX-A reported more than three months’ duration of improvement [19].

In another study of 362 patients, 64.4% of patients receiving abobotulinumtoxinA had improvement with a mean duration of effect of 12.1 weeks and 73.9% of onabotulinumtoxinA patients had improvement with a mean duration of effect of 11.3 weeks [31]. Electromyography-guided abobotulinumtoxinA injections in 13 patients with CD demonstrated a positive moderate correlation between patient-reported and objective improvement in CD symptoms [32]. Similarly, after four consecutive injections, 163 CD patients with a mean dose of 389 U of abobotulinumtoxinA and 44 patients with 145 U of onabotulinumtoxinA reported the duration of treatment effect was 11 ± 1.6 weeks and 10 ± 2.4 weeks, respectively [33]. No statistically significant difference was found between the two treatments.

Mordin et al. (2014) observed significant improvements in SF-36 domain scores with abobotulinumtoxinA compared with placebo for physical functioning, role-physical, bodily pain, general health, and role-emotional domains [14]. In addition, patients treated with abobotulinumtoxinA demonstrated statistically significant improvements versus placebo-treated patients on the VAS for pain severity at weeks 4 and 8. Similarly, another study reported patients treated with abobotulinumtoxinA had statistically significant improvements versus placebo on the VAS for pain severity at weeks 4, 8, and 12 [12]. Poewe et al. (2016) observed significant improvements in health-related quality of life (HRQoL) using the CDIP-58 and VAS scores for pain total score from baseline to week 4 in the abobotulinumtoxinA solution for injection and dry formulation groups compared with placebo [11]. In another study, Lew et al. (2018) observed 23.6% and 38.4% of abobotulinumtoxinA-treated patients rated their CD “much improved” or “very much improved” at weeks 2 and 4, respectively, on the PGIC compared with 6.8% of placebo-treated patients (Table 1) [10]. Although there was no statistically significant difference between abobotulinumtoxinA and placebo in the change from baseline in CDIP-58 total score at week 4, abobotulinumtoxinA demonstrated a statistically significant change from baseline in the CDIP-58 head and neck domain compared with placebo. Van den Bergh et al. (1995) reported 67% of patients had complete pain relief and 25% of patients had >50% pain relief after botulinum toxin injections [21].

Brefel-Courbon et al. (2000) evaluated patients’ global assessments of treatment effect, as well as patients’ HRQoL via the French version of the Nottingham Health Profile (NHP) [26]. More than 60% of patients indicated “moderate” or “marked” improvement for each injection. There was a significant improvement in the pain domain and a numerical improvement in the domains of social isolation and emotional behavior. However, Hefter et al. (2013) observed statistically significant improvement in CDQ-24 total score and five subscales of the CDQ-24 (stigma, emotional well-being, pain, activities of daily living, and social/family life), as well as statistically significant reductions in patient diary item scores (activities of daily living, pain, pain duration) at weeks 4 and 12 [24]. Correspondingly, patient diaries demonstrated statistically significant patient-reported improvements in day-to-day capacities and activities, pain, and duration of pain.

Haussermann et al. (2004) observed that 57 patients (63%) were still being treated with abobotulinumtoxinA after 10‒12 years of follow-up [34]. The primary reasons for discontinuation were inconvenience due to travel and costs, and side effects of therapy (Table 3). In the ANCHOR-CD real-world registry study, 43.6% of patients treated with abobotulinumtoxinA 500 U rated global improvement of change via PGIC at week 4 after cycle 1 injection [28].

### 3.3. Safety

A total of 21 of the 31 identified studies provided safety findings. These are summarized in Table 1, Table 2 and Table 3. The common treatment-emergent adverse events (TEAEs) reported in abobotulinumtoxinA-treated patients in randomized controlled trials (RCTs) were dysphagia, muscular weakness, nasopharyngitis, injection site pain, neck pain, back/shoulder pain, neck muscle weakness, tiredness/fatigue, dry mouth, cold/upper respiratory tract infection, pharyngitis, and headache. A higher rate of TEAEs was observed in patients treated with onabotulinumtoxinA (69%) than with abobotulinumtoxinA (58%), though the difference was not statistically significant (*p* = 0.35) [15]. Dysphagia occurred more frequently with abobotulinumtoxinA than with onabotulinumtoxinA; dysphagia was reported in 16% of patients treated with abobotulinumtoxinA at a 1:3 ratio compared to onabotulinumtoxinA and 17% of patients treated with abobotulinumtoxinA at a 1:4 ratio, compared to 3% of patients treated with onabotulinumtoxinA [17]. Dysphagia did not appear to be a dose- or treatment-cycle-related adverse event [12]. Headache, fatigue, and upper respiratory tract infection occurred less frequently in patients treated with abobotulinumtoxinA [15]. Dysphonia occurred with high-dose (550 U + SD 233) treatment with abobotulinumtoxinA [20].

There were no statistically significant differences between abobotulinumtoxinA and Lanzhou BTX-A regarding occurrence of TEAEs, with dysphagia being the most common TEAE experienced across all treatment sessions in both treatment groups [19]. Botulinum toxin-naïve patients reported more TEAEs (7 of 11 patients; 64%) than did those in the non-naïve group (8 of 24 patients; 33%) [8]. Common TEAEs in observational studies generally mirrored those reported in RCTs.

### 3.4. Health Economic Outcomes

Four studies reported health economic outcomes. In a study by Brefel-Courbon et al. (2000; *N* = 21; Table 3), pharmacoeconomic data were collected from eight months before the first injection of abobotulinumtoxinA through the eight months following the first injection [26]. The mean direct medical cost of CD increased from $97 ± $29 USD per patient/month before treatment (1997 French Franc converted to 1997 USD) to $228 ± $30 USD per patient/month after treatment (*p* < 0.01). The increase in cost after initiating treatment was largely attributed to the cost of abobotulinumtoxinA itself ($157 ± $27 USD per patient/month), because hospital inpatient care costs were null (*p* < 0.01), diagnostic procedure costs were significantly decreased (*p* < 0.05), and other direct medical costs remained relatively similar. Taken together, investigators estimated the annual cost of abobotulinumtoxinA treatment for CD as $2752 ± $360 USD. The investigators concluded that although the cost of treatment may be considered substantial, it may only represent the first year of treatment because the clinical benefits and duration of improvement appear to increase with multiple injections.

In another study of 362 patients seen at four movement disorder clinics in Germany and the United States (Table 3), the annual cost of abobotulinumtoxinA and onabotulinumtoxinA for an individual with CD (converted from 1997 Deutschmarks to 1997 USD) was similar ($2419 ± $1038 and $2790 ± $1161 USD, respectively) [31]. However, results were inconsistent in providing sub-analyses of efficacy, safety, and specific medical costs for abobotulinumtoxinA vs. onabotulinumtoxinA in CD, precluding conclusions regarding the cost-benefit differences between the two preparations.

In a retrospective single-center cost analysis comparing patients who switched from onabotulinumtoxinA to abobotulinumtoxinA, the latter was associated with reduced median toxin reimbursement costs ($1710 vs. $988 USD, *p* < 0.0001), patient out-of-pocket toxin costs in case of copays or coinsurance ($285 vs. $181 USD, *p* < 0.0001), and the cost of unavoidable toxin waste ($165 vs. $148 USD, *p* < 0.0001). The data projected that treating CD with abobotulinumtoxinA approximately every 13 weeks could provide median savings of US $2844 over a period of 1 year [40].

In a cost-effectiveness analysis modeled on data from the RCT by Truong et al. (2010; Table 1) [41], abobotulinumtoxinA was compared with best supportive care (BSC) for the treatment of CD in the United Kingdom. In the base-case scenario, total incremental quality-adjusted life years (QALYs) gained from abobotulinumtoxinA compared with BSC was 0.235 per patient and the total incremental cost was UK £7160 in direct medical costs, corresponding to an incremental cost-effectiveness ratio (ICER) of UK £30,468 per QALY gained. Several alternative scenarios were also presented, with the scenario considering indirect costs due to productivity loss implying cost-savings with abobotulinumtoxinA versus BSC. Compared with the base-case scenario, ICER per QALY gained was lower (>UK £3000) for the scenario considering a 16-week re-injection interval for abobotulinumtoxinA and the scenario considering vial sharing. Interestingly, ICER versus BSC for onabotulinumtoxinA and incobotulinumtoxinA were UK £48,978 and UK £58,554, respectively, per QALY gained in the base-case scenario.

## 4. Discussion

This systematic literature review identified 31 studies reporting on the safety, efficacy, patient-reported outcomes, and economic outcomes of abobotulinumtoxinA for the treatment of adults with CD.

All six of the double-blind, placebo-controlled clinical trials identified demonstrated the efficacy of abobotulinumtoxinA versus placebo in patients with CD [7,8,9,10,11,12]. Relative to placebo, abobotulinumtoxinA demonstrated efficacy in both botulinum toxin-naïve patients and patients previously treated with botulinum toxin. AbobotulinumtoxinA produced significant decreases from baseline in mean TWSTRS (range, −6.0 to −15.6 ± 2.0) and mean post-treatment Tsui (range, 4.0 to 6.5) scores compared with placebo at week 4, with significant improvement sustained to week 12 (TWSTRS range −9.1 ± 1.7; Tsui mean post-treatment score 4.8). The onset of improvement with abobotulinumtoxinA versus placebo was seen as early as week 2 (TWSTRS score −5.4). Repeated administration of abobotulinumtoxinA was associated with symptomatic improvements in short- and long-term TWSTRS total and subscale scores. The recommended dosing of abobotulinumtoxinA for routine treatments is 250–1000 U; however, it was effective across a range of doses, with dose-dependent improvement observed across doses of 500 to 1000 U, but inconsistent dose-response relationships were noted for lower doses (125 U) [16].

AbobotulinumtoxinA was also shown to be effective in patients with CD in all six active-controlled clinical trials. Studies comparing abobotulinumtoxinA and onabotulinumtoxinA showed no statistically significant differences in efficacy supporting non-inferiority of abobotulinumtoxinA (onabotulinumtoxinA vs. abobotulinumtoxinA 1:3 ratio, *p* = 0.02; onabotulinumtoxinA vs. abobotulinumtoxinA 1:4 ratio, *p* = 0.01) [15,17,33]. Whereas Ranoux et al. (2002) [17] proposed that abobotulinumtoxinA (7 days longer with abobotulinumtoxinA 1:3 ratio (*p* = 0.58), 25 days longer with abobotulinumtoxinA 1:4 ratio (*p* = 0.02)) has a longer duration of action than onabotulinumtoxinA, this finding was not substantiated by other studies. Challenges around defining the bioequivalence ratio complicate interpretation of abobotulinumtoxinA and onabotulinumtoxinA comparisons [10,15,17,18]. Recent publications recommend using a conversion of 1:3 U of onabotulinumtoxinA to abobotulinumtoxinA, although conversion ratios of 1:2.5 might be equally safe and effective [15,17,18].

Patient-reported outcome data generally reflected abobotulinumtoxinA-associated improvements in disease severity. AbobotulinumtoxinA treatment was associated with statistically significant and clinically meaningful improvements across several patient-reported outcome measures, including HRQoL, pain severity, symptom severity, and impression of global change in disease (Table 1).

AbobotulinumtoxinA for the treatment of CD was generally safe and well-tolerated, with few reported severe TEAEs. The most commonly reported adverse effects of abobotulinumtoxinA across studies were dysphagia, injection site pain, muscle weakness, fatigue, and dysphonia. These events were usually mild or moderate, transient, and dose-related, and resolved spontaneously without further interventions. The frequency of TEAEs did not change substantially over time.

The cost of treatment of CD is relatively low when other cost factors, such as non-medical costs or intangible costs, are taken into account. The effect of abobotulinumtoxinA on reducing impairment and improving functional health seems to increase in magnitude and duration after the first year of treatment, suggesting a cumulative clinical effect, which could lower the total cost of treatment after the first year. The cost of treatment may be outweighed by the clinical effects and the impact of abobotulinumtoxinA on QoL. Importantly, treatment may lead to overall cost savings by reducing indirect costs due to productivity loss.

Although this systematic literature review used a robust method of identifying, grading, and summarizing evidence of the safety, efficacy, patient-reported outcomes, and economic outcomes of abobotulinumtoxinA in patients with CD, there are some limitations. The heterogeneity of the identified studies precluded meta-analysis, thus limiting the overall interpretability of the findings. In addition, some studies did not report relevant data (such as specific time points); therefore, it was not possible to compare some outcomes across studies. The limited number of patients in some studies also made it difficult to draw firm conclusions. Lastly, there were no studies comparing newer formulations of toxin A (i.e., rimabotulinumtoxinB, daxibotulinum toxin, incobotulinumtoxinA) to abobotulinumtoxinA in CD, leaving a gap in clinical knowledge. Further research is needed in large placebo- and active-controlled trials with robust reporting of study outcomes in order to provide more empirical evidence of comparative efficacy, cost-effectiveness, and dose conversion in patients with CD treated with botulinum toxin A.

## 5. Conclusions

Our systematic review of abobotulinumtoxinA demonstrated that routine use of abobotulinumtoxinA in CD is well-established and effective. At recommended doses, benefits were sustained for up to 8‒12 weeks, with significant improvements in TWSTRS and Tsui scores as well as pain and QoL. However, in extension studies, re-treatment was determined by a clinical need after a minimum of 12 weeks and the median time to re-treatment was 14 weeks (18 weeks for the 75th percentile). Future studies are needed to compare the beneficial effects of other botulinum toxin formulations relative to abobotulinumtoxinA in CD.

## Figures and Tables

**Figure 1 toxins-12-00470-f001:**
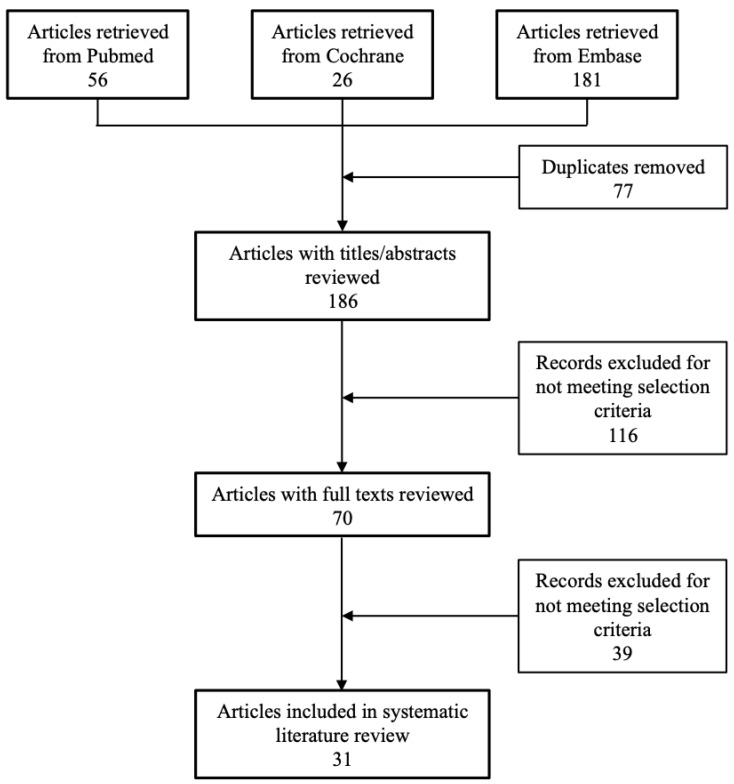
Flow chart of study selection.

**Table 1 toxins-12-00470-t001:** Placebo-controlled studies.

Reference	Patients, n	aboBoNT-A Dose	Efficacy Outcomes	Safety	PROs
Poewe et al., 1998 [7]	aboBoNT-A 250 U, *n* = 19; 500 U, *n* = 16; 1000 U, *n* = 18 Placebo, *n* = 20 (All BoNT-naïve)	250, 500, or 1000 U	Mean modified Tsui score week 4: statistically significant difference vs. placebo for 500 U and 1000 U dose groups (*p* < 0.05)	≥1 TEAE: aboBoNT-A 250 U, 37%; 500 U, 65%; 1000 U, 83%; placebo: 25% Most common TEAEs (250 U, 500 U, 1000 U, placebo, respectively): dysphagia (21%, 29%, 39%, 10%), neck weakness (11%, 12%, 56%, 0%), dry mouth (21%, 18%, 33%, 5%), injection-site discomfort (5%, 18%, 28%, 10%)	Subjective global improvement >50% (placebo, 250 U, 500 U, 1000 U, respectively): week 2: 10%, 11%, 25%, 22% week 4: 10%, 16%, 25%, 44% * week 8: 11%, 11%, 38% *, 50% *
Wissel et al., 2001 [8]	aboBoNT-A, *n* = 35 (BoNT-A-naïve, *n* = 11) Placebo, *n* = 33 (BoNT-A-naïve, *n* = 10)	500 U	Tsui mean score: aboBoNT-A, baseline (mean ± SD): 11.1 ± 1.7; weeks 4 and 8 (adjusted mean ± SEM): 6.5 ± 0.63 and 7.7 ± 0.58, respectively. Placebo, baseline (mean ± SD): 11.5 ± 1.8; weeks 4 and 8 (adjusted mean ±SEM): 9.5 ± 0.67 and 10.1 ± 0.62, respectively. Between-group comparison, week 4: *p* = 0.001; week 8: *p* = 0.002	≥1 TEAE: aboBoNT-A, 43%; placebo: 27% Most common TEAEs: dry mouth, neck muscle weakness, dysphagia, cold, and injection-site pain No serious TEAEs were reported TEAEs were more frequent in BoNT-A treatment–naïve than non-naïve patients: aboBoNT-A group, 7/11 (64%) naïve patients reported 15 TEAEs; 8/24 (33%) non-naïve patients reported 11 TEAEs. Placebo group, 3/10 (30%) naïve patients reported 7 TEAEs; 7/23 (30%) non-naïve patients reported 9 TEAEs	Pain score reduction, OR aboBoNT-A vs. placebo, week 4: 3.3, 95% CI: 1.2 to 9.4, *p* = 0.024; week 8: 2.1, 95% CI: 0.8 to 5.6, *p* = 0.152 Symptom improvement, OR aboBoNT-A vs. placebo, week 4: 8.5, 3.1 to 23.0, *p* < 0.001; week 8: 6.8, 95% CI: 2.5 to 18.3, *p* < 0.001
Truong et al., 2005 [9]	aboBoNT-A, *n* = 37 (BoNT-naïve, *n* = 9) Placebo, *n* = 43 (BoNT-naïve, *n* = 12)	500 U (range 400–500 U)	**Primary**: TWSTRS total score (change from baseline) week 4: aboBoNT-A, −9.9; placebo, −3.8. Difference in adjusted mean changes (ANCOVA): −6.0, 95% CI: −10.6 to −1.3; *p* = 0.013 **Other**: Difference in adjusted mean changes (ANCOVA), 8 weeks: −5.8; 95% CI: −9.9 to −1.6; *p* = 0.007. 12 weeks: −4.3; 95% CI: −8.2 to −0.4; *p* = 0.030	≥1 TEAE: aboBoNT-A, 92%; placebo, 79% Most common TEAEs: neck/shoulder pain (aboBoNT-A, 38%; placebo, 30%), injection-site pain (aboBoNT-A, 38%; placebo, 23%), tiredness (aboBoNT-A, 35%; placebo, 30%), headache (aboBoNT-A, 24%; placebo, 23%), dry mouth (aboBoNT-A, 22%; placebo, 19%), neck muscle weakness (aboBoNT-A, 16%; placebo, 12%), and dysphagia (aboBoNT-A, 16%; placebo, 9%)	Patient-assessed change in SD signs and symptoms, aboBoNT-A vs. placebo, week 4: 15.0; 95% CI: 6.3 to 23.7; *p* < 0.001; week 8: 12.9, 95% CI: 4.9 to 20.9; *p* = 0.002; week 12: 8.4; 95% CI: 1.2 to 15.5, *p* = 0.022 VAS pain score, week 4: −11.4, 95% CI: −21.3 to −1.5, *p* = 0.024; week 8: −8.8, 95% CI: −16.4 to −1.1, *p* = 0.025; week 12: −2.2, 95% CI: −8.3 to 3.9, *p* = 0.480
Lew et al., 2018 [10]	aboBoNT-A, *n* = 89 (BoNT-naïve, *n* = 32) Placebo, *n* = 45 (BoNT-naïve, *n* = 16)	Mean: 451.8 U; median: 500 U	**Primary**: TWSTRS total score, weighted treatment difference from baseline, week 4: −8.3; 95% CI: −12.17 to −4.47 (*p* < 0.001) **Subgroup analysis**: BoNT-naïve vs. previously treated statistically significant differences in TWSTRS total score vs. baseline **Other**: TWSTRS total score, weighted treatment difference vs. baseline at week 2: −5.4; 95% CI: −8.76 to −2.12 (*p* = 0.002)	≥1 TEAE: aboBoNT-A, 40.9%; placebo, 22.2% Serious TEAEs: aboBoNT-A, 4.5%; placebo, 2.2% Most common TEAEs: dysphagia (aboBoNT-A, 9.1%; placebo, 0%), muscular weakness (aboBoNT-A, 9.1%; placebo, 0%), neck pain (aboBoNT-A, 8.0%; placebo, 0%), headache (aboBoNT-A, 5.7%; placebo, 0%)	PGIC, CD rated as “much improved” or “very much improved,” week 4: aboBoNT-A, 38.4%; placebo, 11.1% Week 2: aboBoNT-A, 23.6%; placebo, 6.8% CDIP-58 total score: No statistically significant differences between aboBoNT-A and placebo; head and neck domain: aboBoNT-A, −15.9; placebo, −5.8 (ANOVA *p* = 0.016)
Poewe et al., 2016 [11]	aboBoNT-A solution for injection, n = 156 (BoNT-naïve, *n* = 36); dry formulation, *n* = 159 (BoNT-naïve, *n* = 40) Placebo, *n* = 54 (BoNT-naïve, *n* = 14) (Open-label extension with aboBoNT-A solution for injection, *n* = 333)	500 U	**Primary**: TWSTRS total score (change from baseline), week 4: significant reduction in both aboBoNT-A groups vs. placebo (*p* < 0.0001) (actual values not stated) Non-inferiority of aboBoNT-A solution for injection vs. dry formulation was not established TWSTRS total score, weeks 2 and 8: Significant improvements in both aboBoNT-A groups vs. placebo TWSTRS total score, week 12: Significant improvements in the aboBoNT-A dry formulation group TWSTRS subscale scores, treatment difference vs. baseline, week 4: Significant reduction in both aboBoNT-A groups vs. placebo (*p* < 0.0001) (actual values not reported) **Other**: A post hoc analysis that excluded results from outlier centers (due to heterogeneity) met the non-inferiority criteria Efficacy maintained through open-label phase (up to 4 cycles)	≥1 TEAE, cycle 1: aboBoNT-A solution for injection, 42.5%; dry formulation, 37.8%; placebo: 25.5% Severe TEAEs, cycle 1: aboBoNT-A solution for injection, 3.3%; dry formulation, 5.1%; placebo: 0% Most common TEAEs, cycle 1 (aboBoNT-A solution for injection, dry formulation, placebo, respectively): dysphagia (3.3%, 7.1%, 0%), nasopharyngitis (5.9%, 2.6%, 1.8%), injection-site pain (3.9%, 3.2%, 1.8%), neck pain (2.6%, 3.8%, 1.8%), and headache (3.9%, 1.9%, 1.8%)	CDIP-58 total score, least squares mean reduction, week 4 vs. baseline: aboBoNT-A solution for injection, −9.5; dry formulation, −11.2; placebo, −0.9 (*p* < 0.0001 for active groups vs. placebo) Statistically significant reductions in mean VAS pain scores and symptoms in aboBoNT-A groups vs. placebo. VAS pain score, week 4 vs. baseline: aboBoNT-A solution for injection vs. placebo, *p* < 0.005; dry formulation vs. placebo, *p* < 0.0001. Dry formulation vs. placebo at weeks 8 and 12 vs. placebo: *p* < 0.005 and *p* < 0.05, respectively
Truong et al., 2010 [12] HRQoL analyses (SF-36): Jen et al. 2014 [13]	aboBoNT-A, *n* = 55 (BoNT-naïve, *n* = 10) Placebo, *n* = 61 (BoNT-naïve, *n* = 10)	500 U (open-label extension, mean cycle 1, 502 U; cycle 2, 643 U; cycle 3, 716 U; cycle 4, 776 U)	**Primary**: TWSTRS total score (change from baseline) week 4: (RCT phase), mean (± SEM): aboBoNT-A, −15.6 ± 2.0; placebo, −6.7 ± 2.0 (*p* < 0.001) **Other**: TWSTRS total score (change from baseline), week 12: (RCT phase), mean (± SEM): aboBoNT-A, −9.1 ± 1.7; placebo, −4.9 ± 1.7 (*p* = 0.019) Post hoc efficacy analysis showed no clinically meaningful difference (not defined) for BoNT-naïve vs. non-naïve group **Open-label extension**: Mean changes in TWSTRS total scores from treatment cycle baseline were −16.2, −11.4, −10.8, and −11.3 for cycles 1–4, respectively. (Greater improvements in cycle 1 than subsequent cycles reflect higher treatment cycle baseline scores at cycle 1 than cycles 2–4)	≥1 TEAE: aboBoNT-A, 47%; placebo, 44%. Most TEAEs were mild or moderate Most common TEAE: dysphagia; RCT phase, aboBoNT-A, 9%; placebo, 0%; open-label phase, cycle 1, 12%; cycle 2, 13%; cycle 3, 6%; cycle 4, 10%	SF-36 mental health (change from baseline) week 8, aboBoNT-A vs. placebo: not statistically significant (*p* = 0.061). SF-36 physical health, (change from baseline) week 8, aboBoNT-A vs. placebo: not statistically significant due to hierarchical structure (*p* = 0.002) Statistically significant improvements in aboBoNT-A vs. placebo for VAS pain at weeks 4, 8, and 12 (*p* < 0.001, *p* < 0.001, and *p* = 0.007, respectively) HRQoL analyses (Jen et al.): SF-36, week 8 vs. baseline: No significant differences for vitality, social functioning, or mental health For domain scores, aboBoNT-A was more likely than placebo to show improvement for physical functioning (OR = 1.6; *p* = 0.01), role-emotional (OR = 2.4; *p* = 0.0001), and mental health (OR = 1.5; *p* = 0.007)
Mordin et al., 2014 [14]	Includes only patients who completed the RCT phase: aboBoNT-A, *n* = 45; placebo, *n* = 38)	500 U	Efficacy previously reported in Truong et al. 2010 HRQoL assessed by the SF-36 Health Survey (SF-36) at weeks 0 and 8	Safety previously reported in Truong et al. 2010	HRQoL analyses: SF-36 (change from baseline) week 8: Physical functioning: aboBoNT-A, +8.2; placebo, −1.9 (*p* = 0.018) Role-physical: aboBoNT-A, +16.6; placebo, +3.2 (*p* = 0.008) Bodily pain: aboBoNT-A, +13.9; placebo, +2.9 (*p* = 0.010) General health: aboBoNT-A, +3.2; placebo, −2.5 (*p* = 0.030) Role-emotional: aboBoNT-A, +9.5; placebo, +4.3 (*p* = 0.030)

* *p* < 0.05; aboBoNT-A, abobotulinumtoxinA; ANOVA, analysis of variance; CD, cervical dystonia; CDIP, CD Impact Profile; CI, confidence interval; HRQoL, health-related quality of life, OR, odds ratio; PGIC, Patient Global Impression of Change; PRO, patient-reported outcome; RCT, randomized controlled trial; SD, standard deviation; SEM, standard error of the mean; SF-36, 36-Item Short Form health survey; TEAE, treatment-emergent adverse event; TWSTRS, Toronto Western Spasmodic Torticollis Rating Scale; VAS, visual analog scale.

**Table 2 toxins-12-00470-t002:** Active-controlled studies.

Article	Patients, *n*	Doses	Efficacy Outcomes	Safety	PROs
Odergren et al., 1998 [15]	aboBoNT-A, *n* = 38 onaBoNT-A, *n* = 35	aboBoNT-A, 500 U (mean dose ± SD, 477 U ± 131; range, 240–720) onaBoNT-A, 100 U (mean dose ± SD, 152 U ± 45; range, 70–240)	**Primary:** Tsui score (mean ± SD), week 12 or at re-treatment (if earlier than week 12): aboBoNT-A, 4.8 ± 2.4; onaBoNT-A, 5.8 ± 2.6; difference not statistically significant after adjustment for baseline and center effects Mean time to re-treatment: aboBoNT-A, 84 days (range, 56–122); onaBoNT-A, 81 days (range, 49–111); difference not statistically significant Patients treated prior to week 12: aboBoNT-A, 26%; onaBoNT-A, 32% Patients treated after week 12: aboBoNT-A, 13%; onaBoNT-A, 9%	≥1 TEAE: aboBoNT-A, 58%; onaBoNT-A, 69% Most common TEAEs: dysphagia (aboBoNT-A, 16%; onaBoNT-A, 11%), pharyngitis (aboBoNT-A, 11%; onaBoNT-A, 11%), headache (aboBoNT-A, 8%; onaBoNT-A, 17%), fatigue (aboBoNT-A, 8%; onaBoNT-A, 11%), and upper respiratory tract infection (aboBoNT-A, 8%; onaBoNT-A, 9%) Most TEAEs (>90%) were mild or moderate in severity TEAEs possibly or probably related to study medication, aboBoNT-A: 32% of patients; onaBoNT-A: 26%. Most common = dysphagia	Not reported
Laubis-Hermann et al., 2002 [16]	aboBoNT-A recommended dose, *n* = 15 aboBoNT-A low-dose, *n* = 16	aboBoNT-A recommend-ded dose, 500 MU (mean dose, 547 MU; range, 350–700) aboBoNT-A low-dose, 125 MU (mean dose, 130 MU; range, 63–188)	TWSTRS total score vs. baseline at week 4: aboBoNT-A recommended dose: 5.6 ± 8.1 (*p* < 0.02); aboBoNT-A low-dose: 4.4 ± 5.6 (*p* < 0.01) Subscale scores: aboBoNT-A recommended dose: statistically significant improvements for disability and pain; aboBoNT-A low-dose: statistically significant improvements for disability and severity No statistically significant difference between treatments for the TWSTRS total or subscales scores	Not reported	Improvements in overall CD symptoms, magnitude of maximal effect (up to 5 days following injection): aboBoNT-A recommended dose, “striking improvement”: 21%; “marked response”: 36%; “moderate response”: 29%. aboBoNT-A low-dose, “striking improvement”: 0%; “marked response”: 53%; “moderate response”: 20% Degree of improvement after injection: aboBoNT-A recommended dose, “marked improvement”: 77%; “mild improvement”: 23%. aboBoNT-A low-dose, “marked improvement”: 50%; “moderate improvement”: 29%; “mild improvement”: 14%; “no improvement”: 7%
Ranoux et al., 2002 [17]	Total *N* = 54 (3-period cross-over study with sequential treatment) onaBoNT-A, *n* = 51 aboBoNT-A at a 1:3 onaBoNT-A to aboBoNT-A dose ratio, *n* = 51 aboBoNT-A at a 1:4 onaBoNT-A to aboBoNT-A dose ratio, *n* = 52	onaBoNT-A at the usually effective dose (defined as the dose at which a satisfactory response was achieved in the previous two treatments)	Tsui mean score, week 4: onaBoNT-A, 3.22; aboBoNT-A 1:3 ratio, 4.32; aboBoNT-A 1:4 ratio, 4.89. onaBoNT-A vs. aboBoNT-A 1:3 ratio, *p* = 0.02; onaBoNT-A vs. aboBoNT-A 1:4 ratio, *p* = 0.01 TWSTRS pain score: onaBoNT-A vs. aboBoNT-A 1:3 ratio, *p* = 0.04; onaBoNT-A vs. aboBoNT-A 1:4 ratio, *p* = 0.02 Mean duration of action vs. onaBoNT-A: 7 days longer with aboBoNT-A 1:3 ratio (*p* = 0.58), 25 days longer with aboBoNT-A 1:4 ratio (*p* = 0.02) No significant differences between two aboBoNT-A groups	≥1 TEAE: onaBoNT-A, 18%; aboBoNT-A 1:3 ratio: 33%; aboBoNT-A 1:4 ratio: 36% Most common TEAE: dysphagia (onaBoNT-A, 3%; aboBoNT-A 1:3 ratio: 16%; aboBoNT-A 1:4 ratio: 17%) No TEAE severe enough for study withdrawal	Not reported
Yun et al., 2015 [18]	aboBoNT-A and onaBoNT-A at 2.5:1.0 dose ratio, *n* = 94 (4-week washout period between the 16-week treatment cycles)	aboBoNT-A: 361.04 U ± 657.91 (range, 200–400) onaBoNT-A: 144.41 U ± 623.16 (range, 80–160)	Tsui mean score, week 4: aboBoNT-A, 4.0 ± 3.9; onaBoNT-A, 4.8 ± 4.1 (*p* = 0.091) TWSTRS total and subscale scores: differences not statistically significant	≥1 TEAE: aboBoNT-A, 14.9%; onaBoNT-A, 20.2% Most common TEAEs: neck muscle weakness (aboBoNT-A, 9.6%; onaBoNT-A, 13.8%), dysphagia (aboBoNT-A, 6.4%; onaBoNT-A, 12.8%), and neck/shoulder pain (aboBoNT-A, 2.1%; onaBoNT-A, 7.4%)	Not reported
Barbosa et al., 2015 [19]	aboBoNT-A, *n* = 14 Lanzhou BTX-A, *n* = 20	Equivalency ratio of 3 U of aboBoNT-A per 1 U of Lanzhou BTX-A	**Primary:** TWSTRS total score (change from baseline), week 4: aboBoNT-A, −12.78, 95% CI: −6.68 to −18.88; *p* = 0.001. Lanzhou BTX-A: −9.98; 95% CI: −6.38 to −13.58; *p* <0.001. No statistically significant difference between treatments (*p* = 0.38) **Other:** Statistically significant (*p* < 0.05) decreases from baseline in all TWSTRS subscale scores with both treatments; no statistically significant differences between treatments. Similar results observed for last injection (≤5 injections over 13 months)	Most common TEAEs in both treatment groups: dysphagia (27.3%), injection-site pain (4.5%), muscle weakness (1.3%) No statistically significant differences between aboBoNT-A and Lanzhou BTX-A groups regarding occurrence of TEAEs With the exception of one dysphagia event after the fourth injection, all dysphagia events with aboBoNT-A were mild (*n* = 22)	After first treatment, improvement for >3 months: aboBoNT-A, 14%; Lanzhou BTX-A: 25%. Improvement for 2–3 months: aboBoNT-A, 71%; Lanzhou BTX-A: 45% After fifth treatment, improvement for >3 months: aboBoNT-A, 55%; Lanzhou BTX-A: 50%. Improvement for 2–3 months: aboBoNT-A, 45%; Lanzhou BTX-A: 50%
Bigalke et al., 2001 [20]	aboBoNT-A Group A, *n* = 8; Group B, *n* = 38. Groups differed regarding monitoring and examinations	Group A, first 3× high dose (658 U ± 232), then ≥3× low dose (262 U ± 68) Group B, first 3× high dose (550 U ± 233), then 3× low dose (235 U ± 100)	Group A: Investigator-assessed symptom severity, beginning of relief and duration of improvement deemed as effective with high dose as with low dose Group B: Improvement rating, beginning of relief, and duration of improvement deemed as effective with high dose as with low dose	TEAEs, Group A, high dose: neck weakness, *n* = 4 patients; dysphagia, *n* = 1. Low-dose: none TEAEs, Group B, high-dose: neck weakness, *n* = 21 patients; dysphagia, *n* = 15; pain, *n* = 11; dysphonia, *n* = 4. Low-dose: neck weakness, *n* = 6; dysphagia, *n* = 1; pain, *n* = 4	Not reported

aboBoNT-A, abobotulinumtoxinA; CD, cervical dystonia; onaBoNT-A, onabotulinumtoxinA; PRO, patient-reported outcome; SD, standard deviation; SF-36, 36-item Short Form health questionnaire; TEAE, treatment-emergent adverse event; TWSTRS, Toronto Western Spasmodic Torticollis Rating Scale.

**Table 3 toxins-12-00470-t003:** Other studies.

Article	Study Type; Patients, *n*	Doses	Efficacy Outcomes	Safety	PROs
Van den Bergh et al., 1995 [21]	Open-label study; with CD, *n* = 28	aboBoNT-A, mean ± SD dose per treatment cycle: 384 MU ± 188 (range: 63–1045)	38-point composite score (based on: subjective rating 0–5, a Tsui score, and a video score), before treatment: 18.9 ± 4.4 (range 8.6–26.2); at peak improvement: 5.2 ± 3.0 (range 0–13.3) Last treatment (mean: 5 cycles) vs. first treatment, pre-treatment scores: –40% (*p* < 0.0001); post-treatment scores: –35% (*p* = 0.03)	Mild dysphagia in 2 patients with rotatocollis	Complete pain relief: 67% of patients; >50% pain relief: 25% of patients
Kessler et al., 1999 [22]	Prospective study, *n* = 303	aboBoNT-A, mean per treatment: 778 ± 253 U	Greatest reduction in modified Tsui score after first injection (baseline median, 10; after injection median, 6; change from baseline, ‒3.7). Continued improvement (though less pronounced) through 6 injections. Tsui scores generally consistent after sixth injection, with median Tsui score of 4 before the 15th injection. Reduction in Tsui score statistically significant over first 6 injections (*p* < 0.0001)	≥1 TEAE: 75% of patients TEAEs considered possibly related to treatment: 22% of treatments (685 of 3088 sessions) Most common TEAE: dysphagia (77.1% of all TEAEs), with most events mild or moderate in severity (87%) Other common TEAEs included neck muscle weakness (17.2% of all TEAEs), dry mouth (9.9%), neck pain (4.7%), voice changes (4.2%)	Not reported
Jamieson et al., 1997 [23]	Prospective observational study, *n* = 14	aboBoNT-A, range, at first post-treatment video: 200–1000 U; at second post-treatment visit: 200–1200; highest dose: 400–1500	Treatment duration 4 years 5 months to 6 years 7 months. Statistically significant (*p* < 0.05) difference between early and later post-treatment Tsui scores	Dysphagia: 8/15 patients (53%) treated long term	Not reported
Hefter et al., 2013; 2011 [24,25]	Prospective cohort study; aboBoNT-A, *n* = 503	aboBoNT-A, 500 U	Tsui score, baseline: 8.4 ± 3.5; (change from baseline), week 4: ‒3.83; 95% CI: 4.01–3.57; *p* < 0.0001. Statistically significant improvements in all subscale scores Mean increase in symptom improvement: 44.3% (± 34.8%). Improvement sustained through week 12 of treatment cycle	≥1 TEAE: 41.4% of patients Most common TEAEs: muscular weakness (13.8%), dysphagia (9.9%), and neck pain (6.6%) Most TEAEs (89.7%) were mild or moderate in severity, and 30.1% of TEAEs were considered treatment-related Serious TEAEs: 11 patients (2.1%); 2 considered possibly treatment-related At week 4, 86.7% of investigators and 80.3% of patients rated the tolerability of study medication as “good” or “very good.” Corresponding values at week 12 were 88.8% (investigators) and 85.4% (patients)	CDQ-24 vs. baseline, at week 4: ‒11.1; 95% CI: ‒12.5 to ‒9.6; *p* < 0.001. At week 12: ‒11.8; 95% CI: ‒13.1 to ‒10.4; *p* < 0.001. Improvements in all 5 subscales at weeks 4 and 12 (*p* < 0.001) Statistically significant improvements in patient-reported day-to-day capacities and activities, pain, and duration of pain (rated using an 11-point VAS) at both weeks 4 and 12 (*p* < 0.001) Pain relief (less or no pain), week 4: 66% of patients; week 12: 74% of patients Change from baseline in CDQ-24 total and subscale scores correlated with change from baseline in Tsui total score
Brefel-Courbon et al., 2000 [26]	Prospective; aboBoNT-A, *n* = 21	Mean injected aboBoNT-A dose per session: 450 U ± 18 (range, 160–1120 U)	Tsui score, mean at baseline: 18.4; at 1 month after first injection: 10.5 (*p* < 0.0001) Average of 8.7 weeks of improvement Tsui scores continued to decrease and duration of improvement generally increased through the fifth injection	Most common AEs: dysphagia (45.9%), local pain (25.0%), muscle weakness (12.5%), fatigue (8.3%), and dysphonia (8.3%). None were considered “serious.” None resulted in study dropouts	Patients’ global assessments of treatment effect, after first injection: “marked” improvement, 52%; “moderate” improvement, 33%. For each injection, >60% of patients indicated “moderate” or “marked” improvement NHP: significant improvement in pain domain by end of study (*p* = 0.02); numerical improvements in domains of social isolation and emotional behavior
Misra et al., 2012 [27]	Prospective observational study (interest in CD); *n* = 404	aboBoNT-A, median dose, 500 U (10% received ≥1000 U) onaBoNT-A; median dose, 160 U (10% received ≥300 U)	Treatment responders (defined as ≥25% TWSTRS severity scale improvement at visit 2 or 3 vs. visit 1; ≥12 weeks effect duration; no severe TEAEs; and patient-rated CGI score ≥2 at visits 2 and 3), overall: 28.6%; 95% CI: 24.0 to 33.5. aboBoNT-A ~33% and onaBoNT-A ~23% (actual values not reported)	No significant differences between treatment groups for the percentage of patients with ≥1 TEAE, ≥1 severe TEAE, or dysphagia at visit 2	Not reported
Trosch et al., 2017 [28]	Prospective observational study (ANCHOR-CD registry); *n* = 304	aboBoNT-A, mean dose: 504 U ± 229; median 500 U, range: 100–1500 U	TWSTRS total score decreased by 27.4% (± 28.9) from baseline to week 4, with a 31.7%, 18.5%, and 25.3% decrease in the TWSTRS severity, disability, and pain subscale scores, respectively.	Seventeen patients (5%) reported a total of 39 TEAEs. Of these patients, 2 with dysphagia and 1 with blurred vision and chewing difficulty withdrew from the study Most common TEAEs: dysphagia (*n* = 6, 1.7%), muscular weakness (*n* = 4, 1.2%), neck pain (*n* = 3, 0.9%), and rhinorrhea (*n* = 2, 0.6%) Of the 39 TEAEs, *n* = 31 (79.5%) were assumed related to study drug	Global improvement of change (via PGIC) as “much improved” or “very much improved” at week 4 after cycle 1 injection: 43.6% of patients Global satisfaction subscale of the modified TSQM measure at cycle 1, week 4: 48.7%
Bentivoglio et al., 2017 [29]	Retrospective cohort study; aboBoNT-A, *n* = 39	Mean injected aboBoNT-A dose per session: 701.5 U (median, 560 U; range, 60–1560 U)	Tsui score, before treatment: 5.7 ± 1.8 points (range, 2‒11); maximum efficacy: 3.5 ± 1.5 points (range, 0‒9); *p* < 0.01 ≥2-point reduction in Tsui score in 70.9% of the treatments Mean latency: 6.4 ± 3.0 days (range, 1–30) Mean overall duration of the clinical improvement: 93 days (range, 0–300) Median inter-treatment interval: 131 days (95% CI: 87–191)	Most common TEAEs: posterior neck muscle weakness (15.1%), rigidity (2.7%), dysphagia (1.9%), injection-site pain (1.2%)	VAS, mean score before injection: 4.4 ± 1.8 (range 0–8); at maximum efficacy: 1.8 ± 1.6 (range 0–8); *p* < 0.01 CGA, mean for all treatments: 3.6 ± 1.0 (range 0–6)
Hefter et al., 2014 [30]	Retrospective cohort study, *n* = 568	aboBoNT-A, overall dose not reported; mean dose in group with PSTF: 752 U ± 32; in group without PSTF: 703 U ± 56	PSTF (≥4 Tsui scores collected during treatment with ≥3 consecutive aboBoNT-A injections): 5.8% of patients; estimated incidence: 1.6% per year (or 14.5% over 9 years) Time of onset of PSTF varied (e.g., after 4 injections, after 38 injections)	Not reported	Not reported
Dodel et al., 1997 [31]	Prospective, patients with CD, overall *n* = 362; aboBoNT-A, *n* not reported; onaBoNT-A, *n* not reported	Mean dose per treatment, aboBoNT-A: 732.3 U ± 239.5; onaBoNT-A: 187.3 U ± 68.0	Onset of effect (days), aboBoNT-A: 9.4 ± 7.7; onaBoNT-A: 5.4 ± 3.1 Duration of effect (weeks), aboBoNT-A: 12.1 ± 5.0; onaBoNT-A: 11.3 ± 3.2	Not reported separately for CD; overall AE rate, aboBoNT-A: 26%; onaBoNT-A: 15% (*p* < 0.001)	Patient-rated % response, aboBoNT-A: 64.4 ± 20.1; onaBoNT-A: 73.9 ± 13.6
Finsterer et al., 1997 [32]	Prospective, invasive (EMG), *n* = 13	aboBoNT-A, mean per treatment: 223 U (range: 140–320 U)	Turns/s (T/S), pre-injection: 411 (range, 201–746); post-injection: 289 (range, 101–868); *p* = 0.0001 Amplitude/turn (A/T), pre-injection: 304 µV (range, 137–500); post-injection: 202 µV (range 120–354 µV); *p* = 0.0001	No severe side effects. Local and reversible hematoma in 1 patient	12/13 patients reported a benefit from treatment at 4 weeks (“moderate” improvement, *n* = 4; “slight” improvement, *n* = 8; no change, *n* = 1)
Mohammadi et al., 2009 [33]	Database study, overall *n* = 207; aboBoNT-A, *n* = 163 onaBoNT-A, *n* = 44	aboBoNT-A, mean dose, 389 U ± 144 onaBoNT-A; mean dose, 145 U ± 44	Mean latency to response, aboBoNT-A: 7.6 ± 3.5 days; onaBoNT-A: 7.7 ± 3.3 days Duration of treatment effect, aboBoNT-A: 11 ± 1.6 weeks; onaBoNT-A: 10 ± 2.4 weeks	Most common side effects: neck muscle weakness (aboBoNT-A: 5%; onaBoNT-A: 7%); dysphagia, mild (aboBoNT-A: 8%; onaBoNT-A: 9%); injection-site pain (aboBoNT-A: 9%; onaBoNT-A: 6%) No severe or systemic side effects Secondary non-response, aboBoNT-A: 3.7%; onaBoNT-A: 4.5%	CGI, aboBoNT-A: 2.5 ± 0.3 weeks; onaBoNT-A: 2.2 ± 0.4 weeks; difference not statistically significant
Haussermann et al., 2004 [34]	Longitudinal cohort study, *n* = 90	aboBoNT-A, mean per treatment session: 833 MU ± 339	Secondary non-response in 3/90 patients during follow-up	≥1 TEAE: *n* = 34 patients Most common TEAEs: neck weakness (*n* = 13 patients), mild dysphagia for food (*n* = 12), general weakness (*n* = 5)	>60% of patients were still being treated with aboBoNT-A after ≤12 years. Mean score, global subjective rating of effect (–4 = very bad, +4 = very good): 1.93 ± 1.18
Marchetti et al., 2005 [35]	Retrospective observational study of patients switching, overall *n* = 70; from aboBoNT-A to onaBoNT-A, *n* = 63; from onaBoNT-A to aboBoNT-A, *n* = 7	Mean dose, patients switching from aboBoNT-A to onaBoNT-A: aboBoNT-A, 601 ± 234 U; onaBoNT-A, 130 ± 44 U Mean dose, patients switching from onaBoNT-A to aboBoNT-A: aboBoNT-A, 468 ± 139 U; onaBoNT-A, 112 ± 30 U	Not reported	≥1 TEAE: aboBoNT-A, *n* = 37 patients; onaBoNT-A, *n* = 22 patients Most common TEAEs: dysphagia (aboBoNT-A, *n* = 19 events; onaBoNT-A, *n* = 12 events), followed by flu-like symptoms (aboBoNT-A, *n* = 4 events; onaBoNT-A, *n* = 0 events)	Not reported
Rystedt et al., 2012 [36]	Retrospective study using patient casebook notes, *n* = 75	onaBoNT-A then switch to aboBoNT-A Median dose, last 4 onaBoNT-A injections: 97.5 U (range 40–200); first 4 aboBoNT-A injections: 200 U (range 80–420)	Not reported	≥1 TEAE, last 4 onaBoNT-A treatments: *n* = 4 patients (5.3%; dysphagia, *n* = 2; pain and dizziness, *n* = 2) ≥1 TEAE, first 4 aboBoNT-A treatments: *n* = 18 patients (24%; dysphagia, *n* = 8; neck weakness, *n* = 4; pain, *n* = 4; dizziness, *n* = 2; tremor, *n* = 1; nausea, *n* = 1)	Patients reporting more effective treatment after switching to aboBoNT-A: *n* = 12 (16%) Patients reporting worse effect after switching to aboBoNT-A: *n* = 4 (5%) Patients reporting not feeling any difference at all in effect: *n* = 59 (79%)
Vivancos-Matellano et al., 2012 [37]	Retrospective chart review, *n* = 37	Mean dose: 487 U (range 320–650)	97% of patients maintained treatment response	≥1 TEAE: 8/37 patients treated with aboBoNT-A (9 TEAEs in total) Most common TEAE: dysphagia, *n* = 7 patients (18.9%); typically mild in severity One AE led to study discontinuation	Not reported

aboBoNT-A, abobotulinumtoxinA; AE, adverse event; AT, amplitude/turn; CD, cervical dystonia; CDQ-24, 24-item Craniocervical Dystonia Questionnaire; CGA, clinical global assessment; EMG, electromyography; NHP, Nottingham Health Profile; onaBoNT-A, onabotulinumtoxinA; PGIC, Patient Global Impression of Change; PRO, patient-reported outcome; PSTF, partial secondary treatment failure; P/T, parameter turns; TEAE, treatment-emergent adverse event; SD, standard deviation; TSQM, Treatment Satisfaction Questionnaire for Medication; TWSTRS, Toronto Western Spasmodic Torticollis Rating Scale; VAS, visual analog scale.

## Data Availability

Data sharing is not applicable to this article as no new data were created or analyzed in this study.

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
