# Peer review of "Use of AbobotulinumtoxinA in Adults with Cervical Dystonia: A Systematic Literature Review"

_toxins, 2020, doi:10.3390/toxins12080470_

Round 1

Reviewer 1 Report

This was a systematic literature review on the use of abobotulinumtoxinA in aduts affected of CD.

The aim of the authors was to assess the clinical evidence about the effects of abobotulinumtoxinA for tretment of CD in studies of safety, efficay, patient-reported outcomes, and economic outcomes, by searching in comprehensive electronic medical literature databases . The searching strategy was to use a combination of medical subject heading terms and keywords. Out of a total of 263 publications, only 31 articles met the selection criteria, and eligibility criteria and then were considered for the systematic review. Regarding the clinical efficacy, the patient-reported outcomes and/or the safety data, the data results were obtained by 6 placebo-controlled trials, 6 active-controlled trials, and 16 observational studies. Data on health economic outcomes wre provided in one of the clinical trials, in one specific cost-analysis publication, and in two of the observational studies.

The conclusions were that the routine use of abobotulinumtoxinA in CD is well-established and effective with improvements in TWSTRS and Tsui scores, in pain, and quality of life. The effcicacy was sustained for up 8-12 weeks with a median retreatment time of 14 weeks.

The strenght of this systematic review is given by the number of studies included. However, as also the authors stated by themselves, the main limitation of this review is due to the heterogenicity of the selected studies that precluded a meta-analysis with a consequent limitation in the interpretation of the findings. In addition, some studies did not report relevant data and for this reason, it was not possible to compare some outcomes in the different studies.

In general, this is a complete and well-written systematic review on the use of abobotulinumtoxinA in CD. The review is exhaustive and covers all the most important points of this topic.

Minor points: too much words in the Tables. I’d try to reduce the sentences in the Tables in order to make them more readable

Author Response

Reviewer Comments:

Reviewer #1:  

This was a systematic literature review on the use of abobotulinumtoxinA in adults affected of CD.

The aim of the authors was to assess the clinical evidence about the effects of abobotulinumtoxinA for treatment of CD in studies of safety, efficacy, patient-reported outcomes, and economic outcomes, by searching in comprehensive electronic medical literature databases. The searching strategy was to use a combination of medical subject heading terms and keywords. Out of a total of 263 publications, only 31 articles met the selection criteria, and eligibility criteria and then were considered for the systematic review. Regarding the clinical efficacy, the patient-reported outcomes and/or the safety data, the data results were obtained by 6 placebo-controlled trials, 6 active-controlled trials, and 16 observational studies. Data on health economic outcomes were provided in one of the clinical trials, in one specific cost-analysis publication, and in two of the observational studies.

The conclusions were that the routine use of abobotulinumtoxinA in CD is well-established and effective with improvements in TWSTRS and Tsui scores, in pain, and quality of life. The efficacy was sustained for up 8-12 weeks with a median retreatment time of 14 weeks.

The strength of this systematic review is given by the number of studies included. However, as also the authors stated by themselves, the main limitation of this review is due to the heterogenicity of the selected studies that precluded a meta-analysis with a consequent limitation in the interpretation of the findings. In addition, some studies did not report relevant data and for this reason, it was not possible to compare some outcomes in the different studies.

In general, this is a complete and well-written systematic review on the use of abobotulinumtoxinA in CD. The review is exhaustive and covers all the most important points of this topic.

Minor points: too much words in the Tables. I’d try to reduce the sentences in the Tables in order to make them more readable

Author Response: We thank the Reviewer for nicely summarizing our work and highlighting its major strengths. We appreciate that Tables 1-3 can be overwhelming; however, they are intended to summarize all of the manuscripts and results captured in the SLR and the relevant results from those articles. Some edits have been made to reduce wordiness but ultimately, we feel that the level of detail is necessary to completely inform the reader of the scope of the data.

Edit to the manuscript: Reduced sentences in Tabled 1-3 where possible to ensure better readability.

Reviewer 2 Report

As I whole, I thought that this was a well documented and detailed study that acknowledges the shortcomings of the review in that it is not able to perform a Meta-Analysis with the heterogenous studies.  The method of the review as well as the way that the data is presented in appropriate.

I have no major concerns, but I would like for the authors to provide some more details about how abobotulinum toxin compares to the other botulinum toxins for reduction in TWSTRS scores, length of effect, overall improvement in QoL, etc.  Though they are not able to be directly compared, given that no studies have done this, a quick discussion on the reported benefits of the other botulinum toxins in comparison would be a nice addition to demonstrate rough comparison.

To that end, within the discussion, a brief synopsis of the results of the studies would provide for some more distillation of reported benefits.  The authors report that abobotulinum toxin is effective and provides benefit, but do not help to encapsulate that the benefits are and require the reader to go through each study listed and make that comparison.  Though the studies are not able to be easily compared to one another given the limitations listed, a general statement on percentage improvement reported and the ranges would be helpful.

Author Response

Reviewer Comments:

Reviewer #2:

As I whole, I thought that this was a well-documented and detailed study that acknowledges the shortcomings of the review in that it is not able to perform a Meta-Analysis with the heterogenous studies.  The method of the review as well as the way that the data is presented in appropriate.

I have no major concerns, but I would like for the authors to provide some more details about how abobotulinum toxin compares to the other botulinum toxins for reduction in TWSTRS scores, length of effect, overall improvement in QoL, etc.  Though they are not able to be directly compared, given that no studies have done this, a quick discussion on the reported benefits of the other botulinum toxins in comparison would be a nice addition to demonstrate rough comparison.

To that end, within the discussion, a brief synopsis of the results of the studies would provide for some more distillation of reported benefits.  The authors report that abobotulinum toxin is effective and provides benefit, but do not help to encapsulate that the benefits are and require the reader to go through each study listed and make that comparison.  Though the studies are not able to be easily compared to one another given the limitations listed, a general statement on percentage improvement reported and the ranges would be helpful.

Author Response: We appreciate the comments from Reviewer #2. Unfortunately, direct comparisons are out of scope for this Systematic Literature Review. In addition, we are afraid that commenting on the effect of other products would mislead the readers as the scientific value of any statement in that respect would be highly speculative. I hope that the Reviewer will find this approach reasonable but we are open to reconsider our position should this be unacceptable in their opinion. To help encapsulate the benefits of abobotulinumtoxinA we have agreed to add additional details such as TWSTRS ranges and Tsui mean post-treatment scores to the efficacy statements in the discussion section.

Edit to the manuscript: No edits to have manuscript made regarding comment #1. Changes made in regards to comment #2 include TWSTRS ranges and Tsui score mean post-treatment score ranges into the discussion of the efficacy section, as well as p-values showing possible significance in the duration of treatment with abobotulinumtoxinA.